# Compliance with the WHO recommended 8+ antenatal care contacts schedule among postpartum mothers in eastern Uganda: A cross-sectional study

**Seungwon Lee**[1], **Eminai Adam**[2], **Andrew Marvin Kanyike**[3], **Solomon Wani**[4], **Samuel Kasibante**[5], **David Mukunya**[4,6], **Ritah Nantale**[4,6]*

1 Department of Neuroscience, University of Pennsylvania, Philadelphia, Pennsylvania, United States of America, 2 Department of Research and Innovation, Sanyu Africa Research Institute, Mbale, Uganda, 3 Department of Internal Medicine, Research Associate, Mengo Hospital, Kampala, Uganda, 4 Department of Community and Public Health, Faculty of Health Sciences, Busitema University, Mbale, Uganda, 5 Department of Community Health, Jinja Regional Referral Hospital, Jinja, Uganda, 6 Accelerating Innovations in Maternal, Adolescent, Reproductive and Child Health, Mbale Uganda

* ritahclaire24@gmail.com

**Data Availability Statement:** All relevant data are within the manuscript and its Supporting information files.

## Abstract

### Background

The World Health Organization (WHO) recommends at least 8 antenatal care (ANC) contacts during pregnancy, but many women in low and middle-income countries do not adhere to this schedule, which may contribute to high rates of maternal and neonatal mortality. This study assessed compliance to the WHO recommended 8+ ANC contacts schedule and associated factors among postpartum mothers in eastern Uganda.

### Methods

This was a cross-sectional multicenter study conducted between July and August 2022 at four selected hospitals in Eastern Uganda using quantitative techniques. We recruited post-natal mothers who had given birth within 48 hours with records of their ANC contacts. Compliance to the WHO recommended 8+ ANC contacts schedule was defined as having received the recommended ANC contacts as per the gestational age at childbirth following the current ANC for a positive pregnancy experience WHO guidelines. We conducted multi-variable logistic regression analysis to assess the association between compliance to the WHO recommended 8+ ANC contacts schedule and selected independent variables.

### Results

A total of 1104 postpartum mothers participated in the study with a mean age (± standard deviation) of 26 ± 6.4 years, and a majority had given birth from a referral hospital (n = 624 56.5%). Compliance to the WHO recommended 8+ ANC contacts schedule was low (n = 258, 23.4%), and only 23.2% (196) of the women had attended their first antenatal care contact within the first trimester. Factors associated with compliance to the WHO recommended

**Funding:** The author(s) received no specific funding for this work.

**Competing interests:** The authors have declared that no competing interests exist.

8+ ANC contacts were: attending the first antenatal care contact within 12 weeks of gestation [AOR: 6.42; 95% CI: (4.43 to 9.33)], having 2 to 4 children [AOR: 0.65; 95% CI: (0.44 to 0.94)], having a spouse who is unemployed [AOR: 1.71; 95% CI: (0.53 to1.08)] and having insurance coverage [AOR: 2.31; 95% CI: (1.17 to 4.57)].

## Conclusion

Compliance with the 8+ ANC contacts schedule remains very low. Efforts should focus on increasing health education, particularly for multiparous women, and encouraging mothers to begin ANC in their first trimester. Exploring the dynamics of partner support, especially with employment status, may offer insights into improving ANC attendance.

## Introduction

Globally, approximately 800 mothers die daily due to preventable causes related to pregnancy and childbirth, and 95% of such deaths occur in low- and middle-income countries [1]. Countries with the highest maternal mortality ratios (MMR) are heavily concentrated in sub-Saharan Africa, it accounted for approximately 70% of global maternal deaths in 2020 [1]. Uganda has a maternal mortality ratio of 336 maternal deaths per 100,000 live births and ranks ninth in the world for highest absolute number of maternal deaths [2, 3]. Uganda's progress in reducing MMR is very slow and severely inadequate to fulfill the sustainable development goal target of reducing MMR to less than 70 deaths per 100,000 live births by 2030 [2, 4, 5].

The World Health Organization (WHO) has continuously recommended skilled and timely antenatal care (ANC) as an effective preventive measure against maternal and neonatal deaths in low and middle-income countries [3, 6]. Prior to 2016, the WHO's standard guidelines suggested at least four ANC contacts during pregnancy. With more recent findings that the limited frequency of ANC increases the risk of perinatal deaths, the WHO updated their baseline recommendation to eight ANC contacts during one's pregnancy [7, 8]. Subsequently, the Ministry of Health in Uganda adopted the eight ANC contacts in 2018. An ANC contact refers to an active connection between a pregnant woman and a health care provider that includes provision of preventative medical care to the mother to promote the health of the pregnant mother and child.

However, the uptake and utilization of ANC services in Uganda remains low [9]. A further analysis of the Uganda Health Demographic Survey revealed that 61.1% of mothers aged 15–49 attended at least four ANC contacts while only 1.9% attended eight or more times, as recommended by the WHO [10]. Further studies also show that women in Eastern Uganda were least likely to complete at least eight or more ANC contacts compared to those in the Northern, Western, and Central regions [10]. This further highlights the importance of understanding the variety of socioeconomic, demographic, religious, cultural, maternal and obstetric, and health system related factors to the uptake of ANC services in Eastern Uganda [10–13].

There is a dearth of evidence in understanding the compliance to the WHO recommended 8+ ANC contacts and associated factors in Eastern Uganda, which is critical to inform policy and strengthen health systems to reduce MMR in Uganda. This study, therefore, aimed at assessing compliance and associated factors with the WHO recommended 8+ ANC contacts

in relation to gestation age at delivery among immediate postnatal women at four health facilities in Eastern Uganda.

## Materials and methods

### Study design and setting

This was a hospital-based cross-sectional study using quantitative techniques. The study was conducted in four selected health facilities in Eastern Uganda between July and August 2022. Uganda has a 5-tier public health system with national and regional referral hospitals at the top with specialized services, then general hospitals at the district level, health center IVs (HCIV) at the sub-district level. Sub-district level facilities serve as referral facilities for health center IIIs, IIs and lastly Village Health Teams (VHTs) or community health workers in the community. Antenatal care is normally offered at health center IIIs and above. There are three regional referral hospitals and six general hospitals in eastern Uganda. Four hospitals were purposively selected because they conduct the highest number of deliveries in the region. These include Jinja Regional Referral Hospital (JRRH), Iganga General Hospital (IGH), Mbale Regional Referral Hospital (MRRH), and Kamuli General Hospital (KGH). The selected hospitals have a high patient volume offering comprehensive ANC services to most areas within Eastern Uganda. JRRH is located in Jinja District with a bed capacity of 600 and conducts roughly 500 deliveries per month. The hospital has a functional antenatal care unit in the department of Obstetrics and Gynaecology. The unit receives approximately 280 ANC mothers every day. The hospital has a diagnostic unit that offers radiology services. However, the unit faces the challenge of services interruption primarily due to stock out of commodities. IGH is located in Iganga District, with a bed capacity of 100 and conducts about 550 deliveries per month. MRRH is located in Mbale District, serves as a referral hospital for 13 districts with a bed capacity of 500 and conducts about 800 deliveries per month [14]. The hospital runs ANC services 5 days a week and the ANC clinic receives approximately 330 mothers every day. Like JRRH, MRRH also has diagnostic unit and faces challenges regarding provision of ultra scan services. KGH is located in Kamuli district with a bed capacity of 100 and conducts about 200 deliveries per month. The ANC clinic receives approximately 230 mothers per day. It is a district general hospital with no ultrasound scan services therefore ANC mothers access this service from private providers.

### Study population

We included postnatal mothers in the selected health facilities, who had given birth within 48 hours and gave informed written consent. Mothers who were too sick to communicate and those without a record of their ANC contacts through an ANC card or book during the interview were excluded.

### Sample size estimation and sampling procedure

We assumed a 1.9% prevalence of eight of more ANC contacts, a prevalence obtained from a further analysis of the Uganda Demographic Health Survey 2016 [10]. We considered a precision of 5% and 95% confidence interval and non-response rate of 10% and a design effect of 2. This gave a total sample size of 64 participants. We further calculated a sample size for estimating the factors associated with attending 8+ antenatal care contacts. Maternal age was used to estimate the sample size for factors, using findings from Uganda Demographic Health Survey 2016 where 61.5% of mothers aged 20 to 30 years received four or more ANC contacts, while 52.6% of mothers aged 35 to 49 received four or more ANC contacts. This gave us a maximum

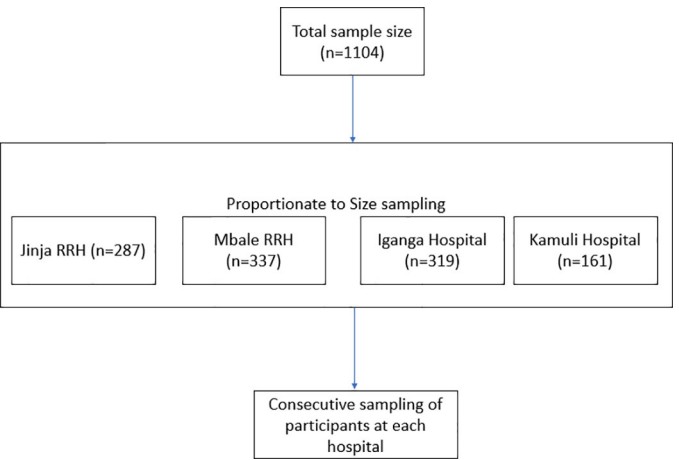

**Fig 1. Schematic presentation of the sampling procedure.**

sample size of 1104 mothers We used proportionate to size sampling basing on the average number of deliveries to estimate the sample size for each hospital (Fig 1). Consecutive sampling was used to recruit the eligible mothers until the required sample size was attained at each hospital. We recruited 337 mothers from Mbale RRH, 287 from Jinja RRH, 319 from Iganga hospital and 161 from Kamuli hospital.

## Study variables

The dependent variable was compliance to the WHO recommended 8+ ANC contacts among mothers. This was defined as having received the recommended ANC contacts as per the gestational age at childbirth according to the current ANC for a positive pregnancy experience WHO guidelines, and not just the total number of contacts, as shown in Table 1. For instance, a mother who delivered at 36 weeks with at least 6 ANC contacts was compliant even if she didn't have 8 contacts. Independent variables included; sociodemographic characteristics (age, maternal and paternal education level, marital status, maternal and paternal occupation, religion, residence, ANC Knowledge, family income, cost of ANC services, place of ANC, distance to health facility, and HIV status) Obstetric characteristics (planned pregnancy, parity, current or previous obstetric complications, history of illness during current pregnancy, first ANC contact timing).

**Table 1. 2016 WHO ANC model.**

| First trimester | |
|---|---|
| Contact 1 | Up to 12 weeks |
| **Second trimester** | |
| Contact 2 | 20 weeks |
| Contact 3 | 26 weeks |
| **Third trimester** | |
| Contact 4 | 30 weeks |
| Contact 5 | 34 weeks |
| Contact 6 | 36 weeks |
| Contact 7 | 38 weeks |
| Contact 8 | 40 weeks |

## Data collection and procedure

Data were collected by eight trained research assistants using a data entry questionnaire designed using Kobo Toolbox (Cambridge, Massachusetts, USA). Kobo Toolbox is an open-source software developed by the Harvard Humanitarian Initiative with support from United Nations Agencies, CISCO, and partners to support data management by researchers and humanitarian organizations (https://www.kobotoolbox.org/). On a daily basis, the research assistants identified the eligible mothers at the postnatal ward, created rapport, and thoroughly explained to them the purpose of the study. Mothers who were interested in taking part in the study signed the consent form and voluntarily participated. All completed questionnaires were uploaded onto Kobo Toolbox servers.

## Statistical analysis

We summarized numerical data as means and standard deviations and categorical data as frequencies and proportions. We conducted multivariable logistic regression to assess factors associated with compliance to the WHO recommended 8+ ANC contacts' schedule. Factors with a p-value less than 0.2 at bivariable analysis, those with biological plausibility and those known to affect compliance to the WHO recommended 8+ ANC contacts schedule from literature were added to the multivariable model. Statistical significance was set at a p-value <0.05. Data were analyzed in Stata version 14.1.

## Ethical approval and consent to participate

The study was conducted according to the Declaration of Helsinki and in line with the principles of Good Clinical Practice and Human Subject Protection. Prior to collecting data, we sought ethical clearance from the Research Ethics Committee of Busitema University Faculty of Health Sciences, approval number BUFHS-2022-4. We also sought administrative clearance from the participating hospitals before conducting the study. Participation in the study was voluntary, and written informed consent was obtained from the participants before data collection. Participant's personal identifiers like home address, identification numbers or mobile numbers were not collected to ensure privacy and confidentiality.

## Results

### Participant characteristics

Overall, 1,104 postpartum mothers participated in the study with a mean age of 26.0 (SD: 6.4) years, mostly within the age group of 20–34 years (n = 805, 72.9%). The majority had delivered from a referral hospital (n = 624 56.5%), were married (n = 903, 81.8%), and from an urban setting (n = 574, 52.0%). Most mothers had attained a secondary/tertiary education level (n = 669, 60.6%) and did not have an insurance cover (n = 1043, 94.5%). The majority of compliant mothers with the 8 ANC contact schedule were aged 20–34 years (n = 200, 77.5%) and gave birth in a general hospital (n = 143, 55.4%). Compliance was also higher among married women (n = 221, 85.7%) compared to single women (n = 37, 14.3%). In addition, mothers who initiated their first ANC visit at or before 12 weeks of pregnancy (n = 162, 62.8%) showed greater compliance than those who started their visits after 12 weeks (Table 2).

### Compliance to the WHO recommended 8+ ANC contacts' schedule

A total of 258 out of 1104 (23.4%) were complaint to the WHO recommended 8+ ANC contacts' schedule and 358 (32.4%) had attended their first antenatal care visit within the first twelve weeks of gestation (first trimester) (Fig 2).

**Table 2. Sociodemographic characteristics of the participants.**

| Variable | Non-compliant to the 8 ANC contacts schedule, n (%) | Compliant to the 8 ANC contacts schedule, n (%) | Total, n (%) |
|---|---|---|---|
| **Study center** | | | |
| Referral hospital | 509 (60.2) | 115 (44.6) | 624 (56.5) |
| General hospital | 337 (39.8) | 143 (55.4) | 480 (43.5) |
| **Maternal age** Mean±SD: 26.0±6.4 years | | | |
| 15–19 | 131 (15.5) | 35 (13.6) | 166 (15) |
| 20–34 | 605 (71.5) | 200 (77.5) | 805 (72.9) |
| ≥35 | 110 (13.0) | 23 (8.9) | 133 (12) |
| **Marital status** | | | |
| Single | 164 (19.4) | 37 (14.3) | 201 (18.2) |
| Married | 682 (80.6) | 221 (85.7) | 903 (81.8) |
| **Religion** | | | |
| Christian | 579 (68.4) | 180 (69.8) | 759 (68.8) |
| Muslim | 267 (31.6) | 78 (30.2) | 345 (31.3) |
| **Highest education level of the mother** | | | |
| None/Primary | 342 (40.4) | 93 (36) | 435 (39.4) |
| Secondary/Tertiary | 504 (59.6) | 165 (64) | 669 (60.6) |
| **Highest education level of the partner/husband** | | | |
| None/Primary | 262 (31) | 58 (22.5) | 320 (29) |
| Secondary/Tertiary | 584 (69) | 200 (77.5) | 784 (71) |
| **Maternal occupation** | | | |
| Employed | 511 (60.4) | 163 (63.2) | 674 (61.1) |
| Unemployed | 335 (39.6) | 95 (36.8) | 430 (38.9) |
| **Paternal occupation** | | | |
| Employed | 711 (84) | 226 (87.6) | 937 (84.9) |
| Unemployed | 135 (16) | 32 (12.4) | 167 (15.1) |
| **Residence** | | | |
| Rural | 410 (48.5) | 120 (46.5) | 530 (48.0) |
| Urban | 436 (51.5) | 138 (53.5) | 574 (52.0) |
| **Monthly income** | | | |
| Less than 100,000 shillings | 340 (40.2) | 95 (36.8) | 435 (39.4) |
| Between 100,000 and 500,000 shillings | 387 (45.7) | 124 (48.1) | 511 (46.3) |
| Above 500,000 shillings | 119 (14.1) | 39 (15.1) | 158 (14.3) |
| **Distance to health facility from place of residence** | | | |
| Less than 1km | 246 (29.1) | 72 (27.9) | 318 (28.8) |
| 1-5km | 347 (41) | 90 (34.9) | 437 (39.6) |
| >5km | 253 (29.9) | 96 (37.2) | 349 (31.6) |
| **Do you have insurance coverage?** | | | |
| Yes | 38 (4.5) | 23 (8.9) | 61 (5.5) |
| No | 808 (95.5) | 235 (91.1) | 1043 (94.5) |
| **Parity**, Mean±SD: 2.7±2.2 | | | |
| Primiparous | 293 (34.6) | 106 (41.1) | 399 (36.1) |
| Multiparous | 397 (46.9) | 128 (49.6) | 525 (47.6) |
| Grand multiparous | 156 (18.4) | 24 (9.3) | 180 (16.3) |
| **Planned pregnancy** | | | |
| Yes | 589 (69.6) | 206 (79.8) | 795 (72) |
| No | 257 (30.4) | 52 (20.2) | 309 (28) |
| **HIV status** | | | |

(*Continued*)

**Table 2.** (Continued)

| Variable | Non-compliant to the 8 ANC contacts schedule, n (%) | Compliant to the 8 ANC contacts schedule, n (%) | Total, n (%) |
|---|---|---|---|
| Positive | 69 (8.2) | 10 (3.9) | 79 (7.2) |
| Negative | 758 (89.6) | 247 (95.7) | 1005 (91) |
| Unknown | 19 (2.2) | 1 (0.4) | 20 (1.8) |
| **Where did you go for antenatal care services?** | | | |
| Public Hospital | 790 (93.4) | 231 (89.5) | 1021 (92.5) |
| Private Hospital | 56 (6.6) | 27 (10.5) | 83 (7.5) |
| **Gestational age at first ANC contact**, Mean±SD: 17.7±8.2 weeks | | | |
| ≤12 weeks | 196 (23.2) | 162 (62.8) | 358 (32.4) |
| >12 weeks | 650 (76.8) | 96 (37.2) | 746 (67.6) |
| **Cost of ANC services** | | | |
| Free | 764 (90.3) | 225 (87.2) | 989 (89.6) |
| Affordable | 80 (9.5) | 30 (11.6) | 110 (10) |
| Unaffordable | 2 (0.2) | 3 (1.2) | 5 (0.5) |
| **Current or previous obstetric complications** | | | |
| Yes | 192 (22.7) | 74 (28.7) | 266 (24.1) |
| No | 654 (77.3) | 184 (71.3) | 838 (75.9) |

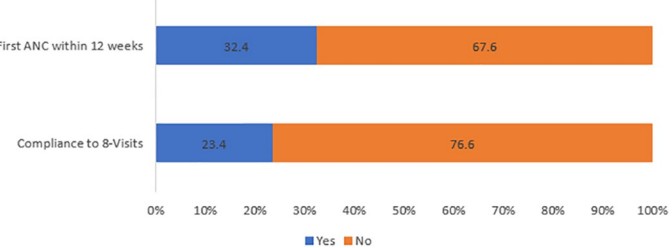

**Fig 2. A graph showing level of compliance to WHO recommended 8 contact ANC schedule and recommended timing for first ANC visit.**

## Factors associated with compliance to the WHO recommended 8+ ANC contacts schedule among immediate postpartum mothers in eastern Uganda

Table 3 shows the factors associated with compliance to the WHO recommended 8+ ANC contacts' schedule. Paternal occupation, insurance, parity and first ANC visit timing were associated with compliance to the WHO recommended 8+ ANC contact's schedule. Mothers who had unemployed partners had 1.71 times the odds of being compliant to the WHO recommended 8+ ANC contacts schedule [AOR: 1.71; 95% CI: (0.53–1.08)]. Mothers with insurance cover had 2.31 times the odds of being compliant to the WHO recommended 8+ ANC contacts schedule [AOR: 2.31; 95% CI: (1.17–4.57)]. Multiparous (2 to 4) women had 35% less odds of being compliant to the WHO recommended 8+ ANC contacts schedule [AOR: 0.65; 95% CI: (0.44–0.94)]. Grand multiparous women (≥5) had 63% less odds of being compliant to the WHO recommended 8+ ANC contacts schedule [AOR: 0.37; 95% CI: (0.19–0.71)]. Mothers who had their first ANC contact within the first trimester had 6.42 times the odds of being compliant to the WHO recommended 8+ ANC contacts schedule [AOR: 6.42; 95% CI: (4.43–9.33)].

**Table 3. Factors associated with compliance to the WHO recommended 8+ ANC contacts schedule among immediate postpartum mothers in eastern Uganda.**

| Variable | COR | 95% CI | P-value | AOR | 95% CI | P-value |
|---|---|---|---|---|---|---|
| **Study center** | | | | | | |
| Referral hospital | 1 | | | | 1 | |
| General hospital | 1.88 | (1.42–2.49) | <0.001 | 1.03 | (0.70–1.52) | 0.865 |
| **Maternal age** | | | | | | |
| 15–19 | 1 | | | | | |
| 20–34 | 1.24 | (0.82–1.86) | 0.304 | 0.95 | (0.55–1.63) | 0.858 |
| 35 years and above | 0.78 | (0.44–1.40) | 0.411 | 1.23 | (0.55–2.76) | 0.614 |
| **Marital status** | | | | | | |
| Single | 1 | | | | 1 | |
| Married | 1.44 | (0.97–2.12) | 0.067 | 0.95 | (0.59–1.53) | 0.836 |
| **Religion** | | | | | | |
| Christian | 1 | | | | 1 | |
| Muslim | 0.94 | (0.69–1.27) | 0.687 | 0.74 | (0.52–1.04) | 0.081 |
| **Maternal education** | | | | | | |
| None/Primary | 1 | | | | 1 | |
| Secondary/Tertiary | 1.20 | (0.90–1.61) | 0.208 | 0.91 | (0.60–1.36) | 0.638 |
| **Paternal education** | | | | | | |
| None/Primary | 1 | | | | 1 | |
| Secondary/Tertiary | 1.55 | (1.12–2.14) | 0.009 | 1.39 | (0.92–2.10) | 0.121 |
| **Maternal occupation** | | | | | | |
| Employed | 1 | | | | 1 | |
| Unemployed | 0.89 | (0.67–1.19) | 0.424 | 0.75 | (0.53–1.08) | 0.122 |
| **Paternal occupation** | | | | | | |
| Employed | 1 | | | | 1 | |
| Unemployed | 0.75 | (0.49–1.13) | 0.164 | 1.71 | (1.01–2.87) | 0.044 |
| **Residence** | | | | | | |
| Rural | 1 | | | | 1 | |
| Urban | 1.08 | (0.82–1.43) | 0.583 | 1.00 | (0.68–1.45) | 0.988 |
| **Monthly income** | | | | | | |
| less than 100,000 shillings | 1 | | | | 1 | |
| Between 100,000 and 500, 000 shillings | 1.15 | (0.85–1.55) | 0.378 | 0.89 | (0.59–1.33) | 0.558 |
| Above 500, 000 shillings | 1.17 | (0.77–1.80) | 0.464 | 1.12 | (0.62–2.01) | 0.710 |
| **Distance to health facility** | | | | | | |
| less than 1km | 1 | | | | 1 | |
| 1-5km | 0.89 | (0.62–1.26) | 0.499 | 1.03 | (0.69–1.53) | 0.894 |
| >5km | 1.30 | (0.91–1.84) | 0.149 | 1.40 | (0.91–2.15) | 0.123 |
| **Insurance coverage** | | | | | | |
| Yes | 2.08 | (1.21–3.56) | 0.008 | 2.31 | (1.17–4.57) | 0.015 |
| No | 1 | | | 1 | | |
| **Parity** | | | | | | |
| 1 | 1 | | | | 1 | |
| 2 to 4 | 0.89 | (0.66–1.20) | 0.449 | 0.65 | (0.44–0.94) | 0.022 |
| > = 5 | 0.43 | (0.26–0.69) | 0.001 | 0.37 | (0.19–0.71) | 0.003 |
| **Planned pregnancy** | | | | | | |
| Yes | 1 | | | | 1 | |
| No | 0.58 | (0.41–0.81) | 0.001 | 0.70 | (0.47–1.05) | 0.089 |
| **HIV status** | | | | | | |

(*Continued*)

**Table 3.** (Continued)

| Variable | COR | 95% CI | P-value | AOR | 95% CI | P-value |
|---|---|---|---|---|---|---|
| Positive | 1 | | | 1 | | |
| Negative | 2.25 | (1.14–4.43) | 0.019 | 1.36 | (0.64–2.89) | 0.421 |
| Unknown | 0.36 | (0.04–3.02) | 0.348 | 0.24 | (0.03–2.22) | 0.212 |
| **Where did you go for ANC** | | | | | | |
| Public hospital | 1 | | | | 1 | |
| Private hospital | 1.65 | (1.02–2.67) | 0.042 | 1.01 | (0.47–2.21) | 0.970 |
| **First ANC contact within 12 weeks** | | | | | | |
| Yes | 5.60 | (4.15–7.54) | <0.001 | 6.42 | (4.43–9.33) | <0.001 |
| No | 1 | | | 1 | | |
| **Cost of ANC** | | | | | | |
| Free | 1 | | | | 1 | |
| Affordable | 1.27 | (0.82–1.99) | 0.287 | 1.38 | (0.67–2.83) | 0.377 |
| Unaffordable | 5.09 | (0.85–30.67) | 0.076 | 8.43 | (0.95–7.92) | 0.056 |
| **Current or previous complications** | | | | | | |
| Yes | 1 | | | | 1 | |
| No | 0.73 | (0.53–1.00) | 0.050 | 0.73 | (0.51–1.04) | 0.083 |

## Discussion

We found out that compliance to the WHO-recommended 8 or more ANC contacts schedule was low, only 23.4% of the women were compliant. This finding is higher than the 1.9% compliance rate reported in the 2016 Uganda Health Demographics survey [10], this could be explained by the fact that, the survey utilized data collected before the endorsement of the WHO eight ANC guideline and the findings cannot be credited to the implementation or non-implementation of the guideline. It could be that before 2016, women who had eight or more ANC contacts were probably high-risk pregnancies that required close monitoring by a healthcare provider. However, our findings are similar to several other low and middle-income countries as many fall short of their ANC targets. A study from Nigeria revealed a compliance rate of 20% for 8 or more ANC contacts [11]. On the other hand, a much lower compliance rate was reported in a Cameroonian study where only 9% of women attained a minimum of 8 contacts [15]. Likewise, a study among 8 sub-Saharan African countries reported a pooled compliance of 7.7% for 8 or more ANC contacts [16]. Countries in Southeast Asia also presented very low compliance rates, with Myanmar and Bangladesh presenting only 18% and 6%, respectively, of pregnant women attending at least 8 contacts [12, 17].

The differences in the compliance rates could be attributed to the method used to determine compliance in the different studies; in this study, we calculated compliance based on meeting the expected number of contacts by gestational age at delivery as per the WHO-recommended schedule, unlike just counting total number of contacts used in other studies. We deem this approach more objective and could explain the relatively higher rate reported in our study. Additionally, differences in socio-economic status and access to health could be the reason for the variations in compliance rates across different regions.

Our study found that 32.4% of mothers had their first ANC visit within the recommended first 12 weeks, which is lower than the rates reported in Myanmar and Ethiopia, where 47% and 33.7% of women began ANC by 12 weeks, respectively [12, 13]. Women who attended their first ANC visit within the first 12 weeks of pregnancy had 6.4 times the odds of complying

with the eight or more recommended contacts than those who did not. This is supported by previous research which has linked early initiation of ANC contacts in the first trimester with the higher frequency and quality of antenatal care contacts [18]. For instance, women who had late booking for ANC, were less likely to have 8 or more ANC contacts compared to women who initiated ANC contacts within the first trimester [19]. Similarly, a study in Ghana showed that for every week increase in gestational age at the time of booking their first ANC contacts at clinics, respondents were less likely to complete all four antenatal contacts at the recommended times [20]. While our study did not examine the factors influencing ANC initiation, previous research has linked timely initiation to literacy, attending public health facilities, being cared for by a doctor or midwife, wealth, and intended pregnancies. Conversely, late initiation is often predicted by multiparity and living in rural areas [21, 22]. Notably, early ANC initiation has been associated with a reduced risk of miscarriages [23]. Early ANC contacts provide an opportunity for early detection and treatment of potential health problems, foster a strong relationship between the woman and her healthcare provider [8]. Women who initiate their ANC contacts early are likely to attend more times because they probably establish a relationship with their healthcare provider, are motivated and more likely to be aware of the importance of regular antenatal care contacts.

Women with parity of greater than or equal to 2 had lower odds of attending the recommended eight ANC visit schedule. This result is in line with what has been reported in other sub-Saharan countries [16, 23]. There may be several reasons for this finding, such as previous positive experiences with pregnancy and childbirth that lead to a false sense of confidence and a belief that frequent prenatal contacts are not necessary. In addition, these women may face obstacles in attending prenatal appointments, such as having limited time due to caring for other children or financial difficulties in providing for a large family. It is worth mentioning that this overconfidence in a positive labor outcome associated with multiparity has also been linked to decreased use of skilled birth attendance services [24, 25]. Targeted interventions could focus on educating multiparous women about the importance of ANC contacts, even in subsequent pregnancies.

The study also revealed that women with health insurance had twice the odds of complying with the recommended eight or more ANC contacts compared to their counterparts. This finding is consistent with other studies that have reported a positive association between health insurance coverage and ANC utilization [26, 27]. The availability of health insurance can reduce the direct out-of-pocket financial burden facilitating regular attendance of antenatal contacts. Women with health insurance may also have greater access to healthcare facilities and may be more likely to receive health education and counseling on the importance of ANC contacts [26]. Expanding access to affordable health insurance could improve ANC attendance and overall maternal health outcomes.

This study identified that women whose partners were unemployed had 1.7 times the odds of attending 8 or more ANC contacts. This is contrary to other studies that have associated partner unemployment with low ANC utilization and late attendance [28, 29]. It's possible that since the study was carried out at public hospitals that offer free healthcare, the financial burden of ANC services may have been less significant for unemployed partners, therefore not a barrier to their women's attendance of ANC contacts. Alternatively, the unemployed partners could have had a leverage to support their wives with home duties while they sought ANC services. Previous research indicates that despite the significance of male involvement in ANC care, demands and restrictions from work are a major barrier to male involvement [30]. Offering flexible ANC services, such as evening and weekend clinics, and workplace health education for employed partners to actively support their partners' ANC visits. Future studies should assess the impact of this approach on ANC attendance.

## Strengths and limitations

This is a multi-center, cross-sectional study that was conducted across major referral facilities in eastern Uganda at various levels of healthcare involving a large number of participants which provides a comprehensive representation of the study population. We determined compliance to the WHO recommended 8+ ANC contacts schedule by evaluating whether the mother followed the recommended contacts according to her gestational age at the time of birth, rather than just the total number of contacts. This increased the objectivity of the study. To minimize recall bias and subjectivity, the study targeted mothers who had recently given birth and had their ANC card information available. However, due to its cross-sectional design, the study cannot establish a causal relationship between the outcome and identified exposures. Further research through longitudinal studies may provide more insights. Additionally, a mixed methods study that combines qualitative in-depth exploration with the quantitative findings could offer a deeper understanding of how identified factors impact compliance.

## Conclusions

Despite the importance of ANC in preventing maternal and child deaths, the study highlights that the compliance to the recommended 8 or more contacts is still low.

When designing targeted interventions to improve adherence to the WHO-recommended 8+ ANC contacts, healthcare providers and policymakers should prioritize enhancing health education, particularly for multiparous women. Efforts should also focus on encouraging mothers to begin ANC visits in the first trimester and offering flexible ANC services that accommodate the involvement of employed partners.

## Supporting information

**S1 File. Inclusivity in global research.**
(PDF)

**S2 File. Study questionnaire.**
(PDF)

**S1 Data.**
(XLSX)

## Acknowledgments

We would like to acknowledge our dear research assistants who were very helpful in data collection. Special regards also go to the study participants for willingly taking time to provide the information and data needed to complete this study.

## Author Contributions

**Conceptualization:** Seungwon Lee, Eminai Adam, Andrew Marvin Kanyike, Solomon Wani, Samuel Kasibante, David Mukunya, Ritah Nantale.

**Data curation:** Seungwon Lee, Eminai Adam, Andrew Marvin Kanyike, Solomon Wani, Samuel Kasibante, David Mukunya, Ritah Nantale.

**Formal analysis:** Eminai Adam, Andrew Marvin Kanyike, Samuel Kasibante, Ritah Nantale.

**Funding acquisition:** Seungwon Lee.

**Investigation:** Seungwon Lee, Eminai Adam, Andrew Marvin Kanyike, Solomon Wani, Samuel Kasibante, David Mukunya, Ritah Nantale.

**Methodology:** Seungwon Lee, Eminai Adam, Andrew Marvin Kanyike, Solomon Wani, Samuel Kasibante, David Mukunya, Ritah Nantale.

**Project administration:** Seungwon Lee, David Mukunya, Ritah Nantale.

**Resources:** Seungwon Lee, Ritah Nantale.

**Software:** Seungwon Lee, Eminai Adam, Andrew Marvin Kanyike, Ritah Nantale.

**Supervision:** Seungwon Lee, David Mukunya, Ritah Nantale.

**Validation:** Seungwon Lee, Eminai Adam, Andrew Marvin Kanyike, Solomon Wani, Samuel Kasibante, David Mukunya, Ritah Nantale.

**Visualization:** Eminai Adam, Andrew Marvin Kanyike, Ritah Nantale.

**Writing – original draft:** Seungwon Lee, Eminai Adam, Andrew Marvin Kanyike, Solomon Wani, Samuel Kasibante, David Mukunya, Ritah Nantale.

**Writing – review & editing:** Seungwon Lee, Eminai Adam, Andrew Marvin Kanyike, Solomon Wani, Samuel Kasibante, David Mukunya, Ritah Nantale.

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
