## [Decision Letter · Decision Letter 0]

26 Dec 2023

PONE-D-23-14122Compliance with the WHO recommended 8+ antenatal care contacts schedule among postpartum mothers in eastern Uganda : A multicenter cross-sectional studyPLOS ONE

Dear Dr. Nantale,

Thank you for submitting your manuscript to PLOS ONE. After careful consideration, we feel that it has merit but does not fully meet PLOS ONE’s publication criteria as it currently stands. Therefore, we invite you to submit a revised version of the manuscript that addresses the points raised during the review process.

We look forward to receiving your revised manuscript.

Kind regards,

Abera Mersha, MSc.

Academic Editor

PLOS ONE

Journal Requirements:

2. Please include the following request in the decision letter, and ping me with follow up. “Please include a complete copy of PLOS’ questionnaire on inclusivity in global research in your revised manuscript. Our policy for research in this area aims to improve transparency in the reporting of research performed outside of researchers’ own country or community. The policy applies to researchers who have travelled to a different country to conduct research, research with Indigenous populations or their lands, and research on cultural artefacts. The questionnaire can also be requested at the journal’s discretion for any other submissions, even if these conditions are not met.  Please find more information on the policy and a link to download a blank copy of the questionnaire here: https://journals.plos.org/plosone/s/best-practices-in-research-reporting. Please upload a completed version of your questionnaire as Supporting Information when you resubmit your manuscript.

3. In the online submission form, you indicated that [The data underlying the results presented in the study are available from the corresponding author upon request].

4. Please include a caption for figure 1.

Reviewers' comments:

Reviewer's Responses to Questions

**Comments to the Author**

1. Is the manuscript technically sound, and do the data support the conclusions?

Reviewer #1: Yes

Reviewer #2: Partly

2. Has the statistical analysis been performed appropriately and rigorously? 

Reviewer #1: Yes

Reviewer #2: Yes

3. Have the authors made all data underlying the findings in their manuscript fully available?

Reviewer #1: No

Reviewer #2: Yes

4. Is the manuscript presented in an intelligible fashion and written in standard English?

Reviewer #1: Yes

Reviewer #2: Yes

5. Review Comments to the Author

Reviewer #1: Introduction

1. Lines 49-50, revise the 1st statement "Globally, approximately 810 mothers die daily due to preventable causes related to pregnancy and childbirth, and 99% of such deaths occur in low- and middle-income countries" to capture the updated or current contribution of maternal deaths from LMIC which is slightly lower that what is stated.

2. Same comment as above about sub-Saharan Africa contribution. Check WHO website on updated figures.

3. Line 68-69 statement "Further studies also show that women in Eastern Uganda were least likely to complete

at least eight or more ANC visits compared to those in the Northern, Western, and Central regions." should be referenced

4. The authors should differentiate ANC visits from ANC contacts and have them defined or described in the introduction and maintain the consistency throughout the text.

Methods

5. What was the criteria for the selection of the four hospitals?

6. The settings description does not mention anything about antenatal care services, numbers of women at these facilities, eg staffing levels, services available like ultrasound scan that attacks women for early ANC etc

7. Why did the study exclude women without antenatal records. Since they were not too sick, self report on the number of ANC contacts would be obtained by interview. What is the effect of excluding them?

8. Sample size calculation for factors associated considered age of the mothers and they obtained sample size 0f 1104. Was clustering taken into account and what were the assumptions

9. The obstetrics variables in lines 125 & 126, includes first ANC contact timing. This is included in the dependent variable. It should be excluded as independent variable. However if they are looking for the number of ANC visits, then it is valid to include ANC timing as independent variable

Results

10. The authors need to revise this statement "Majority had delivered from a referral hospital (n=624 56.5%)". Three four facilities (study sites) with exception of Iganga hospital were regional referrals by the government of Uganda. Re-analyse this.

11. Lines 161-162 the authors state that 196 (23.2%) had attended their first antenatal care visit within the first twelve weeks of gestation. However, table 2 shows higher number of women (total of 358). Please clarify.

12. Line 166, the first ANC visit timing were associated with outcome is misleading (see comment #9). this is part of the measure of the outcome. Should be dropped

13. Line 167-176 is repetition of the results on table 3. consider re-writing that section

14. The variable " where did you go for ANC" appears in the results table 3 first time (not mentioned in the methods). Please clarify.

15. line 196-200. I think the authors have misinterpreted this. Using a measure of recommended schedule by gestation would get fewer women meeting that schedule compared to obtaining 8 visits.

16. I proposed you include as a limitation on the exclusion of women without ANC cards

17. How many of these facilities had an ultrasound scan and which of them was free

Reviewer #2: 1. How long the normal postnatal mothers stay in the hospital before discharge? Is it standard to stay normal labour mothers 48hrs (3days) in the hospital in Uganda? This is because you recruited postnatal mothers who had given birth within 48 hours.

2. Many countries still using old WHO four ANC visits and they are not implementing eight ANC contacts. Does this eight ANC contact schedule is implemented in Uganda? When implemented? Implementation status is important before studying compliance.

3. The measurement of dependant variable is not clear or clearly stated. When a mother compliant to the WHO recommended 8+ ANC contacts? With all 8+ ANC and appropriate gestational age?

4. It is better to show a schematic presentation of the sampling procedure to clarify how the participants are selected from each hospital.

5. You have to be consistent with using ‘ANC contacts’ rather than ‘ANC visits’ when you are talking about new WHO 8+ ANC model because they are quite different.

6. Make your tables quality tables by formatting them to be easily understandable.

7. can you add more "discussion “to your section with research done globally in new WHO 8+ contacts instead of comparing each result with other studies done on old four ANC visit approaches.

8. Please suggest a strong recommendation based on your findings.

6. PLOS authors have the option to publish the peer review history of their article (what does this mean?). If published, this will include your full peer review and any attached files.

Reviewer #1: **Yes: **Sam Ononge

Reviewer #2: No

---

## [Author Response · Author response to Decision Letter 0]

1 Mar 2024

Reviewer’s comment Response to comment Line number

Reviewer 1

Introduction 

1. Lines 49-50, revise the 1st statement "Globally, approximately 810 mothers die daily due to preventable causes related to pregnancy and childbirth, and 99% of such deaths occur in low- and middle-income countries" to capture the updated or current contribution of maternal deaths from LMIC which is slightly lower that what is stated.

 This has been updated according to the latest update from the World Health Organization website. Thank you for this observation. 49-50

2. Same comment as above about sub-Saharan Africa contribution. Check WHO website on updated figures.

 Thank you. We have also updated the sub-Saharan Africa contribution according to the most recent update on the World Health Organization website 51-52

3. Line 68-69 statement "Further studies also show that women in Eastern Uganda were least likely to complete

at least eight or more ANC visits compared to those in the Northern, Western, and Central regions." should be referenced Thank you for this observation and recommendation. This statement has been referenced. 73

4. The authors should differentiate ANC visits from ANC contacts and have them defined or described in the introduction and maintain the consistency throughout the text. The term ‘ANC visits’ have been revised to ‘ANC contacts’ to be more in line with the WHO’s use of vocabulary. Furthermore, we have included a description of what ‘ANC contact’ refers to in the introduction of the manuscript. 

 65-66

Methods 

5. What was the criteria for the selection of the four hospitals? The selected hospitals have a high patient volume offering comprehensive ANC services to most areas within Eastern Uganda. 95-97

6. The settings description does not mention anything about antenatal care services, numbers of women at these facilities, e.g. staffing levels, services available like ultrasound scan that attacks women for early ANC etc. Thank you, we added this to the settings description. 95-109

7. Why did the study exclude women without antenatal records. Since they were not too sick, self-report on the number of ANC contacts would be obtained by interview. What is the effect of excluding them? We excluded women without ANC cards because of recall bias. With a culture of poor antenatal attendance and high illiteracy from prior knowledge in our study group, most women are less likely to precisely remember the times they attended and what was done for them per visit which we would easily ascertain on the card. We believe that we collected large enough sample of those that had cards therefore their exclusion would not significantly affect our results. N/A

8. Sample size calculation for factors associated considered age of the mothers and they obtained sample size of 1104. Was clustering taken into account and what were the assumptions Thank you for your comment. After calculating the sample size using the formula to determining sample size for factors, clustering was taken into account. We identified the average number of deliveries at each of the four hospitals. This gave us the total sample from which we are to get from our calculated sample. The sample size per hospital was obtained by calculating; Number of deliveries per month/ Total sample population* Sample size

We have further highlighted this in the methods section. 131-132

9. The obstetrics variables in lines 125 & 126, includes first ANC contact timing. This is included in the dependent variable. It should be excluded as independent variable. However, if they are looking for the number of ANC visits, then it is valid to include ANC timing as independent variable Thank you for bringing up this important point. Yes, when assessing compliance, we followed the WHO schedule outlined in table 1, line 127. We categorized a woman as compliant if she had the recommended number of contacts at that specific gestational age, regardless of whether her first contact occurred within the first trimester or later. For instance, if a woman is expected to have had four contacts by 30 weeks of pregnancy and she indeed had them, irrespective of when the first one occurred, she would be considered compliant according to our approach. Therefore, having the first ANC visit within the first 12 weeks is not explicitly part of the dependent variable; therefore we treated it as an independent variable. We welcome further discussion on this matter. Thank you for your input. N/A

Results 

10. The authors need to revise this statement "Majority had delivered from a referral hospital (n=624 56.5%)". Three four facilities (study sites) with exception of Iganga hospital were regional referrals by the government of Uganda. Re-analyze this. We had two referral hospitals (Jinja Regional Referral Hospital, and Mbale Regional Referral Hospital) and two general hospitals (Iganga General Hospital (IGH), and Kamuli General Hospital) contrary to what you observed. We therefore think the statement has no issue unless otherwise clarified. Thank you 94-95

11. Lines 161-162 the authors state that 196 (23.2%) had attended their first antenatal care visit within the first twelve weeks of gestation. However, table 2 shows higher number of women (total of 358). Please clarify. Thank you for your observation, this has been clarified. 180

12. Line 166, the first ANC visit timing were associated with outcome is misleading (see comment #9). this is part of the measure of the outcome. Should be dropped Thank you for your comment. Yes, when assessing compliance, we followed the WHO schedule outlined in table 1, line 127. We categorized a woman as compliant if she had the recommended number of contacts at that specific gestational age, regardless of whether her first contact occurred within the first trimester or later. For instance, if a woman is expected to have had four contacts by 30 weeks of pregnancy and she indeed had them, irrespective of when the first one occurred, she would be considered compliant according to our approach. Therefore, having the first ANC visit within the first 12 weeks is not explicitly part of the dependent variable; therefore, we treated it as an independent variable. And its observed there are mothers were compliant to the schedule (had the recommended number of ANC contacts as per gestational age) but didn’t have their first ANC visit in the first 12 weeks). N/A

13. Line 167-176 is repetition of the results on table 3. consider re-writing that section

 We described the results that we thought were salient from table 3 as opposed to rewriting out the entire table results, which we think should be acceptable in scientific writing. Thank you. N/A

14. The variable " where did you go for ANC" appears in the results table 3 first time (not mentioned in the methods). Please clarify. This has been included within the independent variables list in line 142. Thanks for the observation. 142

15. Line 196-200. I think the authors have misinterpreted this. Using a measure of recommended schedule by gestation would get fewer women meeting that schedule compared to obtaining 8 visits. We appreciate your concern. While the frequency of ANC visits is significant, the timing of these visits is equally vital. Failing to attend during the appropriate periods may result in missing critical conditions that could have been managed more effectively and with less impact on pregnancy outcomes. Hence, we believe it's essential for women not only to meet the required number of visits but also to do so within the appropriate time frames. Thank you for your consideration. N/A

16. I proposed you include as a limitation on the exclusion of women without ANC cards. Based on our explanation to response 7 above, we think this wouldn’t be a limitation. N/A

17. How many of these facilities had an ultrasound scan and which of them was free. Thank you, we have clarified this in the settings description. Three of these facilities including Iganga hospital, Jinja and Mbale regional referral hospitals have an ultrasound scan which is free. 95-109

Reviewer 2

 1. How long the normal postnatal mothers stay in the hospital before discharge? Is it standard to stay normal labor mothers 48hrs (3days) in the hospital in Uganda? This is because you recruited postnatal mothers who had given birth within 48 hours. Normal postnatal mothers stay in hospital on average for 24 hours. Mothers were approached before discharge, and those that consented, were recruited into the study. Some mothers were discharged earlier than 24 hours and others were discharged after 24 hours. However, all mothers were recruited before 48 hours postpartum. N/A

2. Many countries still using old WHO four ANC visits and they are not implementing eight ANC contacts. Does this eight ANC contact schedule is implemented in Uganda? When implemented? Implementation status is important before studying compliance. The Uganda Ministry of Health has integrated an eight-contact model into the national sexual and reproductive health policy guideline. In 2018, efforts were initiated to transition towards this eight-contact ANC model. While many facilities have adopted it, the extent of its implementation hasn't been extensively studied. It's a noteworthy observation, and we will take it into account for future research recommendations. Thank you for bringing this to our attention. 63-64

3. The measurement of dependent variable is not clear or clearly stated. When a mother compliant to the WHO recommended 8+ ANC contacts? With all 8+ ANC and appropriate gestational age? Thank you for this comment. We categorized a mother as compliant if the number of contacts she had at a specific gestational age matched the expected visits according to the WHO 8+ contact schedule. These visits might not have occurred precisely at the exact gestational age but within the expected total by that particular gestational age. For example, by the 34th week of gestation, a mother should have had 5 contact visits. If she indeed had these five visits, she would be considered compliant, even if her first visit was at 20 weeks, followed by the third to fifth contacts between weeks 29 to 34, rather than the third visit occurring by week 26 as exactly stated in the schedule. Essentially, our formula correlated the number of contacts with gestational age rather than focusing on specific time stamps for when the ANC visits occurred. N/A

4. It is better to show a schematic presentation of the sampling procedure to clarify how the participants are selected from each hospital. Thank you, we have clarified on the number of participants recruited from each hospital. 133-135

5. You have to be consistent with using ‘ANC contacts’ rather than ‘ANC visits’ when you are talking about new WHO 8+ ANC model because they are quite different. Thank you, this has been done. N/A

6. Make your tables quality tables by formatting them to be easily understandable. 

In light of benchmarking from various studies, we have found that the format in which our tables are presented is quite clear and follows conventional standards. Could you please specify exactly what aspects of the table format you find unclear or problematic? This will help us better understand your concerns and make any necessary improvements. Thank you N/A

7. can you add more "discussion “to your section with research done globally in new WHO 8+ contacts instead of comparing each result with other studies done on old four ANC visit approaches. Thank you, we revised the discussion accordingly.

 203-239

8. Please suggest a strong recommendation based on your findings. The recommendation was included in the conclusion. Second paragraph 292-300

---

## [Decision Letter · Decision Letter 1]

9 Sep 2024

PONE-D-23-14122R1Compliance with the WHO recommended 8+ antenatal care contacts schedule among postpartum mothers in eastern Uganda : A multicenter cross-sectional studyPLOS ONE

Dear Dr. Nantale,

Thank you for submitting your manuscript to PLOS ONE. After careful consideration, we feel that it has merit but does not fully meet PLOS ONE’s publication criteria as it currently stands. Therefore, we invite you to submit a revised version of the manuscript that addresses the points raised during the review process.

Thank you for addressing the initial comments provided by the reviewers. However, major revisions are required before being acceptable for publication. In your revision, please consider my suggestions as well as the suggestions provided by Reviewer 3 for further consideration. 

We look forward to receiving your revised manuscript.

Kind regards,

Muhammad Haroon Stanikzai

Academic Editor

PLOS ONE

Additional Editor Comments:

1- Please proofread the entire article for language correction and grammar.

2- Table 2: Add a single space between the number and frequency in brackets. 164(19.4), it should be corrected as 164 (19.4). Please follow this rule in all tables and text. Line 184: 358 (32.4%)had, it should be 358 (32.4%) had. Please see all text and tables for this suggestion.

3- While discussing timely initiation of ANC, I suggest the authors use, consult, and add the following citation into the revised manuscript (https://www.dovepress.com/sociodemographic-predictors-of-initiating-antenatal-care-visits-by-pre-peer-reviewed-fulltext-article-IJWH).

4- Discussion needs a bit work. The authors should add recommendations at the end of each paragraph from each of the key observation they have made.

Reviewers' comments:

Reviewer's Responses to Questions

**Comments to the Author**

1. If the authors have adequately addressed your comments raised in a previous round of review and you feel that this manuscript is now acceptable for publication, you may indicate that here to bypass the “Comments to the Author” section, enter your conflict of interest statement in the “Confidential to Editor” section, and submit your "Accept" recommendation.

Reviewer #2: All comments have been addressed

Reviewer #3: (No Response)

2. Is the manuscript technically sound, and do the data support the conclusions?

Reviewer #2: Yes

Reviewer #3: Partly

3. Has the statistical analysis been performed appropriately and rigorously? 

Reviewer #2: Yes

Reviewer #3: Yes

4. Have the authors made all data underlying the findings in their manuscript fully available?

Reviewer #2: Yes

Reviewer #3: No

5. Is the manuscript presented in an intelligible fashion and written in standard English?

Reviewer #2: Yes

Reviewer #3: Yes

6. Review Comments to the Author

Reviewer #2: They addressed all my previous comments, and the manuscript is improved. Concerning table quality, my suggestion is that it is better to use a light shading design than a table grid.

Reviewer #3: Comments on” Compliance with the WHO recommended 8+ antenatal care contacts schedule among postpartum mothers in eastern Uganda : A multicenter cross-sectional study”

Overall, this is an interesting and valuable study contributes to the present literature and sharing with a wider audience.

I have not been part of the original review but having read the current version the authors have addressed the concern raised in the previous round.

I have some additional points that need to be addressed for the manuscript to be suitable for publication. Some of the issues remain and others are listed below for the revision.

- The study is trying to address area of a public health important. However, the contents are not coherent and some of the areas are not well described.

- The author used goal –oriented ANC visit and WHO recommended 8+ contacts alternatively. Goal- oriented ANC visited (Focused ANC) is the model used before the recommendation of WHO 8+ ANC contact (ANC for a positive pregnancy experience).It has to be standardized and the consistency should be maintained.

-Introduction and objectives of the study did not the operational definition given for dependent variable. The objective and the introduction section presented about compliance with WHO recommended 8+ contact and associated factors. The definition described the evaluation of compliance of the WHO 8+ recommendation against gestational age. eg. if a woman had her first visit at 34 weeks, how this will be considered in the context of the current study? Make it consistent.

-Please avoid using “didn’t”. write it as “Did not”

-Some of the results needs to presented the details to give clear picture of the findings like compliance of WHO 8+ recommendation against gestational age.

Detailed comments f the manuscripts was attached.

7. PLOS authors have the option to publish the peer review history of their article (what does this mean?). If published, this will include your full peer review and any attached files.

Reviewer #2: No

Reviewer #3: No

---

## [Author Response · Author response to Decision Letter 1]

5 Nov 2024

COMMENT RESPONSE Line number

EDITOR 

1. Please proofread the entire article for language correction and grammar. • We have proofread the manuscript; thank you. N/A

2. Table 2: Add a single space between the number and frequency in brackets. 164(19.4), it should be corrected as 164 (19.4). Please follow this rule in all tables and text. Line 184: 358 (32.4%)had, it should be 358 (32.4%) had. Please see all text and tables for this suggestion. • Thanks for this observation; we have rectified it throughout the manuscript. 189-190

3. While discussing timely initiation of ANC, I suggest the authors use, consult, and add the following citation into the revised manuscript (https://www.dovepress.com/sociodemographic-predictors-of-initiating-antenatal-care-visits-by-pre-peer-reviewed-fulltext-article-IJWH). • Thank you for suggesting the article. We have used information about predictors of timely ANC initiation from this article in our discussion, and it was helpful. 254-258

4. Discussion needs a bit work. The authors should add recommendations at the end of each paragraph from each of the key observation they have made. • We have provided necessary recommendations for our key observations. Thank you 220-297

REVIEWER 3 

General Comments 

5. The author used goal –oriented ANC visit and WHO recommended 8+ contacts alternatively. Goal- oriented ANC visited (Focused ANC) is the model used before the recommendation of WHO 8+ ANC contact (ANC for a positive pregnancy experience). It has to be standardized and the consistency should be maintained. • This has been harmonized throughout the manuscript, thanks for noting that discrepancy and clarification. N/A

6. Introduction and objectives of the study did not the operational definition given for dependent variable. The objective and the introduction section presented about compliance with WHO recommended 8+ contact and associated factors. The definition described the evaluation of compliance of the WHO 8+ recommendation against gestational age. eg. if a woman had her first visit at 34 weeks, how this will be considered in the context of the current study? Make it consistent. • To clarify why we used this approach and its pragmatism. We have added a detailed explanation to study variable sections with an example to give the reader the reason behind this approach. We hope that this can provide more clarity.

• We have added a statement to the study's objective in the introduction to contextualize the definition of our dependent variable. 135 to 138

7. Please avoid using “didn’t”. write it as “Did not” • This has been rectified, thank you. 181

8. Some of the results needs to presented the details to give clear picture of the findings like compliance of WHO 8+ recommendation against gestational age. • We think now with a clear description of our approach in the methods, the reader should be able to appropriately look at the results as presented. Thank you 173-215

9. Line 1-2. The term “ Multcenter “ study for the title may not fulfil the concept of multicentre study. A multicenter study should cover different geographic areas where different languages and cultures practiced. One of its characteristics is about getting ethical clearance at different sites and data transferring across boards. The study was done in one region in eastern Uganda that covers multiple sites not multicenters. The author has change the term to the appropriate one. • We have forfeited the use of the word multicenter in our title. Thank you. 1-2

Abstract 

10. Line 28-29, The current WHO ANC guide line is not goal-oriented. It is called as or "Antenatal care for a positive pregnancy experience". It needs revision • This has been rectified; thanks for noting that discrepancy. 28-29

11. Line 42, Male involvement did not associated with Compliance of WHO 8+ recommendation as it was written in this section. The author should revise it based on the finding of the study and specify the target groups than recommend for all males. • Indeed, the statement regarding partner involvement was general and unclear, not truly representative of the findings. We have completely restructured the conclusion to align with the study findings. 41-44

Methods 

12. Line 82-83, “This was a multi-center cross sectional study involving mainly quantitative methods of data collection”. this section needs to be revised to ensure it is clear for the case of “multicentre” study and if any other methods of data collection used rather than quantitative. • The study design section has been edited for grammatical clarity. We have foregone the use of ‘multicenter’ based on your guidance that doesn’t meet its use and also clarified that this wasn’t just ‘mainly’ but entirely quantitative. 85

13. Ine 87-91, Overall, how many referral and general hospitals found in eastern Uganda? There are three regional referral hospitals and six general hospitals in eastern Uganda 91-93

14. here Line 92-98, what sampling method was used to select those four hospitals rather than delivery load? These hospitals were purposively selected based on the delivery load. 93

15. Did stated method based on delivery load was used for sample size calculation or for selection of study participants using systematic method? We used proportionate to size sampling to estimate the sample size for each hospital and participants at each hospital were recruited by consecutive sampling. 125-126

16. Line 101-102, the author explained that some mothers were discharged before 48hrs of postpartum and the study included those who stay in the facility for 24 hours. If staying for 48 hours after delivery is not a standard in Uganda, in this case the one who stayed more might be those who had complications during either pregnancy or delivery that may enforce them to have more frequent ANC contacts than others. How selection bias was managed in this study? Did the author done any thing to check this bias? All mothers including those discharged before 48 hours were enrolled into study thus selection bias was unlikely. 

17. - Line 115-116, The way the sample size calculated was not clear. The way sample size was calculated was somewhat stringent. Did sample size was calculated at the beginning for all hospitals and the calculated sample size is allocated for each hospitals using proportion to size (PPS) on the number of deliveries or it was calculated independently? If it was calculated, independently how design effect be considered and effect of clusters be managed? The author has to describe it clearly and how show sample size was calculated. Thank you, we have revised this as guided. 125-127

18. Line 116, Did this sampling techniques is probability sampling that make the result of the study be generalized for a wider population? Use a schematic presentation of the sampling procedure to for easy understanding. Thank you, we have added a schematic presentation of the sampling procedure. 129-130

19. Line 121, was it positive pregnancy experiences model? Did this definition implies for all contacts or for a single contacts? Did compliance consider frequency of contact against recommended gestational age or a contact with correct gestational age? We have revised this, thank you. 135

20. Line 134, (https://www.kobotoolbox.org/),should the details not be put under References? Thank you, we added this to the references as suggested 

Results 

21. Line 150-154, the description of in the text did not include the contents of the table. The table contains sociodemographic charterer with the status of compliance of ANC contacts. Describe it as well • We have described that part of the results as well. Thank you. 181-186

22. Line 159-162, this section is the main research question addressed in this study. The frequency with gestational should be presented in detail. Again check the title on Line 159 and definition of dependent variable given in this study to make it consistent. ANC contact Vs gestational age or contact Vs schedule based on the earlier suggestions? • Thank you for your feedback. However, we would like to clarify the study's methodology. Our approach did not simply involve counting ANC visits, but rather evaluating them in relation to gestational age at delivery, which provides a more objective measure. As explained in the methods section, a woman delivering at 36 weeks with 6 contacts would still be on track with the recommended schedule, even though she hadn’t reached 8 visits. This is why we used this method to assess compliance with the "8+ ANC contact schedule" during analysis. At the stage of presenting results, our focus is on reporting on compliance and the factors associated. We don’t see the need again to drag the frequency of visits used in calculating the compliance into the results section. We hope this clears up any confusion 

23. Line 169, the result of regression did not indicated association with compliance of WHO recommendation. Revise the sentence based on the results. • Based on our analysis, “attendance of the first antenatal care contact within 12 weeks of gestation [AOR: 6.42; 95% CI: (4.43 to 9.33)], having 2 to 4 children [AOR: 0.65; 95% CI: (0.44 to 0.94)], having an unemployed spouse [AOR: 1.71; 95% CI: (0.53 to1.08)] and having insurance coverage were associated with compliance. 189-200

24. Table 3, why participants attended their ANC in public and private hospitals were included? What happened to those mothers who attended at health center III? • Participants included in this study delivered at a public hospital, however, we asked them “Where they received ANC services from?” where some reported a public hospital and others a private hospital. If a mother mentioned a health center III, it was a denoted as a public or private hospital accordingly. N/A

25. Line 183 – 186, was the time first ANC contact considered as dependent variable in method section? Figure 1 is not clear and self-explanatory, It is better to present it “X” axis for gestational and schedule of ANC contact and “Y” axis for proportion of mothers than present one. • First ANC visit wasn’t considered as a dependent variable, however we decided to graphically represent it here for its importance in the subject matter of compliance.

• The variables gestation age and contacts per schedule helped us create an objective composite variable—compliance—which is our focus for this paper. That was a means to determine our outcome of interest. N/A

26. 

Discussion 

27. Line 219-221, it is good not to state evidences as for example. Avoid to use such prefixes. Rather synthesize the sentence. • This has been rectified. Thank you for that guidance. 

28. Line 253-256, the sentence did not alien with the objective of the study. The objective of the study was not evaluation of ANC contacts against gestational age. It is evaluation of compliance and associated factors. Revise it accordingly. • We have modified the statement. Although it also meant to provide a probable reason for the differences from a methodological point of view. 

29. Line 260-262, Move this sentence to recommendation sections as a recommendation of future study • This has been done, thank you.

---

## [Decision Letter · Decision Letter 2]

18 Nov 2024

Compliance with the WHO recommended 8+ antenatal care contacts schedule among postpartum mothers in eastern Uganda : A cross-sectional study

PONE-D-23-14122R2

Dear Dr. Nantale,

We’re pleased to inform you that your manuscript has been judged scientifically suitable for publication and will be formally accepted for publication once it meets all outstanding technical requirements.

Kind regards,

Muhammad Haroon Stanikzai

Academic Editor

PLOS ONE

Additional Editor Comments (optional):

A reviewer and myself found your responses and edits to the manuscript to be adequate and recommend acceptance for publication.

Reviewers' comments:

Reviewer's Responses to Questions

**Comments to the Author**

1. If the authors have adequately addressed your comments raised in a previous round of review and you feel that this manuscript is now acceptable for publication, you may indicate that here to bypass the “Comments to the Author” section, enter your conflict of interest statement in the “Confidential to Editor” section, and submit your "Accept" recommendation.

Reviewer #3: All comments have been addressed

2. Is the manuscript technically sound, and do the data support the conclusions?

Reviewer #3: Yes

3. Has the statistical analysis been performed appropriately and rigorously? 

Reviewer #3: Yes

4. Have the authors made all data underlying the findings in their manuscript fully available?

Reviewer #3: Yes

5. Is the manuscript presented in an intelligible fashion and written in standard English?

Reviewer #3: (No Response)

6. Review Comments to the Author

Reviewer #3: All the comments are well addressed. Thank you for revising the manuscript with recommended updates. The author has addressed all concerns raised during the review. The study setting, methods of sample size determination and sampling procedure was made clear in this version. Use of terminologies and interpretation of results of the study were well described and presented in detail. The manuscript is well structured and organized in this version. Those issues that need more description and additional points to make the sentences understandable and clear were explained and discussed very well. The conclusion and recommendations given in the first version were not based on the data rather it presents the facts known so far. In this version the author revised it was based on the data and finding of the study. There is do not have any additional comments.

7. PLOS authors have the option to publish the peer review history of their article (what does this mean?). If published, this will include your full peer review and any attached files.

Reviewer #3: No

---

## [Editor Report · Acceptance letter]

28 Nov 2024

PONE-D-23-14122R2 

PLOS ONE

Dear Dr. Nantale, 

I'm pleased to inform you that your manuscript has been deemed suitable for publication in PLOS ONE. Congratulations! Your manuscript is now being handed over to our production team.

Kind regards, 

on behalf of

Dr. Muhammad Haroon Stanikzai 

Academic Editor

PLOS ONE